# Prediction of Tea Polyphenols, Free Amino Acids and Caffeine Content in Tea Leaves during Wilting and Fermentation Using Hyperspectral Imaging

**DOI:** 10.3390/foods11162537

**Published:** 2022-08-22

**Authors:** Yilin Mao, He Li, Yu Wang, Kai Fan, Yujie Song, Xiao Han, Jie Zhang, Shibo Ding, Dapeng Song, Hui Wang, Zhaotang Ding

**Affiliations:** 1Tea Research Institute, Qingdao Agricultural University, Qingdao 266109, China; 2Tea Research Institute, Rizhao Academy of Agricultural Sciences, Rizhao 276800, China; 3Tea Research Institute, Shandong Academy of Agricultural Sciences, Jinan 250100, China

**Keywords:** tea plant, hyperspectral imaging, machine learning, withering degree, fermentation degree, quality composition

## Abstract

The withering and fermentation degrees are the key parameters to measure the processing technology of black tea. The traditional methods to judge the degree of withering and fermentation are time-consuming and inefficient. Here, a monitoring model of the biochemical components of tea leaves based on hyperspectral imaging technology was established to quantitatively judge the withering and fermentation degrees of fresh tea leaves. Hyperspectral imaging technology was used to obtain the spectral data during the withering and fermentation of the raw materials. The successive projections algorithm (SPA), competitive adaptive reweighted sampling (CARS), and uninformative variable elimination (UVE) are used to select the characteristic bands. Combined with the support vector machine (SVM), random forest (RF), and partial least square (PLS) methods, the monitoring models of the tea polyphenols (TPs), free amino acids (FAA) and caffeine (CAF) contents were established. The results show that: (1) CARS performs the best among the three feature band selection methods, and PLS performs the best among the three machine learning models; (2) the optimal models for predicting the content of the TPs, FAA, and CAF are CARS-PLS, SPA-PLS, and CARS-PLS, respectively, and the coefficient of determination of the prediction set is 0.91, 0.88, and 0.81, respectively; and (3) the best models for quantitatively judging the withering and fermentation degrees are FAA-SPA-PLS and TPs-CARS-PLS, respectively. The model proposed in this study can improve the monitoring efficiency of the biochemical components of tea leaves and provide a basis for the intelligent judgment of the withering and fermentation degrees in the process of black tea processing.

## 1. Introduction

Black tea originated in China [1]. Because it has the characteristics of red leaves, red infusion, sweet and mellow taste, as well as its rich antioxidants, it is loved by people all over the world. Black tea is processed from the fresh leaves of tea tree, including four processes: withering, rolling, fermentation, and drying [2]. Its quality, to some extent, depends on the processing technology [3]. Withering is the first process of black tea processing. When the degree of withering is moderate, it can effectively improve the enzyme system activity in fresh leaves, so as to lay a decent foundation for the subsequent processing and tea quality. Fermentation is the key process of black tea processing [4]. After moderate fermentation, it can promote a series of biochemical reactions centered on polyphenolase oxidation, then change the content of the biochemical components such as tea polyphenols (TPs), free amino acids (FAA), and caffeine (CAF), and finally form the unique flavor quality of black tea. Therefore, monitoring the content of the quality components in the tea withering and fermentation processes is not only one of the important methods to judge the degree of withering and fermentation but also the basis for evaluating the quality of finished black tea. Traditionally, tea makers observe the changes in the tea color and aroma through subjective and empirical methods to judge the withering and fermentation degrees of the tea raw materials. However, this method requires considerable time, manpower, and expertise. The evaluation results are easily affected by the professional and emotional factors of the tea makers, and there are no strict standards. In addition, the content of the quality components determined by a biochemical analysis can also be used to judge the degree of withering and fermentation. For example, the CAF content is determined by ultraviolet spectrophotometry, and the phenolic components and the FAA content are determined by high-performance liquid chromatography [5,6,7]. However, these chemical analysis methods consume samples and time, so they cannot meet the requirements of modern production and monitoring systems. Therefore, developing a fast and accurate method to judge the withering and fermentation degrees of fresh tea leaves is an urgent problem to be solved in the quality control of black tea processing.

Several scholars have developed new methods and technologies to assess tea processing and achieved many results. For example, in 2019, Xu et al. used an electronic nose, electronic tongue, and electronic eye as alternative detection methods to qualitatively evaluate the quality of tea and quantitatively predict the contents of the amino acids, catechins, polyphenols, caffeine, and other chemical components [8]. Although the electronic nose, electronic tongue, and other rapid detection technologies have been widely used to detect the fermentation process of black tea, the design of these instruments is complex, so the measurement results are vulnerable to environmental changes [9,10]. In recent years, spectral analysis technology has been widely used in the determination of biochemical components and the quality analysis of tea leaves [11]. In 2013, Ren et al. used near-infrared spectroscopy combined with a PLS algorithm to determine the main chemical components in black tea, such as CAF, TPs, and FAAs [12]. In 2015, Li et al. used infrared spectroscopy to determine the TPs content in 14 cultivars of tea, which proved the feasibility of infrared spectroscopy to determine the TPs content in tea [13]. In 2015, Diniz et al. used near-infrared spectroscopy combined with PLS and SPA algorithms to determine the TPs and moisture content in tea samples [14]. In 2022, Li et al. constructed a black tea fermentation quality evaluation model by using an ultraviolet–visible spectrum and machine learning algorithm and quantitatively predicted catechin and theaflavin. Among them, the CARS-PLS model has the best performance in evaluating catechin, and the correlation coefficient (R) is as high as 0.91 [15]. Although the above monitoring methods can quickly evaluate the quality of tea, the spectral range is small. When the sample size is large or the test environment is unstable, it will undoubtedly reduce the speed and accuracy of the model. Therefore, there is still a gap between the results achieved by these technologies and people’s expectations, and it is necessary to explore a real-time, fast, accurate, and nondestructive method to solve these problems [16,17,18].

As a novel nondestructive testing technology, hyperspectral imaging technology is developing rapidly [19,20]. It has the characteristics of a high spectral resolution, wide spectral range, and continuous band and has attracted the attention of many researchers [21,22,23]. The results of monitoring the biochemical components of tea by hyperspectral imaging technology have been also reported. For example, in 2021, Ye et al. used hyperspectral images to estimate the non-galloyl and galloyl types of catechins in new shoots of green tea, and the determination coefficient (R^2^) of the estimation model can exceed 0.79 [24]. Yang et al. established a model to quantitatively predict the main endoplasmic components of Congou black tea under a different fermentation time series [25]. Dong et al. applied hyperspectral technology with the chemometrics method to predict the catechin content of tea leaves at different fermentation times [26]. In 2013 and 2014, Xie et al. realized the real-time detection of the color and water content of tea leaves during drying by using hyperspectral image technology [27,28]. At present, there are few reports on the application of hyperspectral imaging technology in black tea processing. However, hyperspectral imaging technology was rarely used to monitor the contents of TP_S_, FAA, and CAF in tea withering and fermentation. In addition, the impact of different band selection methods and modeling algorithms on the performance of tea leaf component content prediction models are rarely reported.

In this study, a hyperspectral imaging system was used to collect the hyperspectral data during the withering and fermentation of fresh tea leaves, and the content of the TPs, FAA, and CAF in each sample was determined. Savitzky–Golay (S-G), multiple scatter correction (MSC), and first derivative (1D) algorithms are used to preprocess hyperspectral data. The monitoring models of the TPs, FAA, and CAF content are constructed through machine learning and various algorithms, which can quantitatively predict the quality components of fresh tea leaves in the withering and fermentation processes and realize the effective discrimination of withering and fermentation degrees. This study lays a good foundation for the nondestructive on-line detection of quality components in the process of tea withering and fermentation and provides a new method for the intelligent judgment of tea withering and fermentation. In addition, the samples in this study are tea-leaves in the two key processing processes of withering and fermentation, which has strong practical significance.

## 2. Materials and Methods

### 2.1. Experimental Design

The experiment was conducted in Rizhao Tea Science Research Institute, Shandong Province, China (119°33′ E, 35°40′ N), on 30 September 2021. The raw materials of fresh tea leaves belonged to the varieties of Zhongcha, and the tenderness was one bud and one leaf. During the withering process of fresh tea leaves, samples were taken once every hour, a total of 19 times. When withering to 16 h, take a large number of withered leaves for rolling, and then enter the fermentation process. During the fermentation process, samples were taken once every 0.5 h, a total of 10 times. For each sampling, the hyperspectral camera (Gaia field Pro-V10, Dualix Spectral Imaging, Chengdu, China) was used to collect the spectral data, and then the samples were put into the oven to dry, sealed, and stored under the condition of −4 °C and dark. In this experiment, each sample was repeated 4 times. A total of 76 and 40 samples were collected during withering and fermentation, respectively; in total, 116 samples were collected in 2 days.

### 2.2. Data Acquisition

#### 2.2.1. Determination of the Contents of TPs, FAA, and CAF

The contents of TPs, FAA, and CAF in the samples were determined and analyzed according to Chinese standards such as GB/T 8313-2002 (determination of tea polyphenols), GB/T 8314-2013 (determination of total free amino acids), and GB/T 8312-2013 (determination of caffeine). The specific methods are as follows:

Preparation of test solution (TS): A total of 1.5 g of tea powder was placed in a 250 mL cup to which 150 mL of water that had been boiled was added, then put into boiling water bath for 45 min. After that, it was filtered immediately while it was hot, the filtrate was transferred into a 500 mL bottle, fixed the volume, and shaken well.

Determination of TPs: A total of 1 mL of TS was transferred to a 25 mL volumetric flask, and 4 mL of water and 5 mL of ferrous tartrate solution were added. After full mixing, phosphate buffer (pH 7.5) was added to the scale, and the absorbance (*A*_1_) was measured at 540 nm.

Determination of FAA: A total of 1 mL of TS was transferred to a 25 mL volumetric flask, 0.5 mL of phosphate buffer (pH 8.0) and 0.5 mL of 2% ninhydrin solution was added, then put into boiling water bath for 15 min. After cooling, water was added to the scale, and the absorbance (*A*_2_) was measured at 570 nm.

Determination of CAF: An amount of 10 mL of TS was transferred to a 100 mL volumetric flask, 4 mL of 0.01 mol/mL hydrochloric acid and 1 mL of lead basic acetate solution was added, water was added to the scale, then it was left to settle and filter. Then, 25 mL of filtrate was transferred to a 50 mL volumetric flask, 0.1 mL of 4.5 mol/L sulfuric acid solution was added, water was added to the scale. After obtaining new filtrate, the absorbance (*A*_3_) was measured at 274 nm.

Preparation of standard curve: The standard curves of FAA and CAF are prepared according to the methods of GB/T 8314-2013 and GB/T 8312-2013, respectively, and the standard regression equation and R^2^ are calculated. Calculation of result: The contents of TPs, FAA, and CAF in tea leaves are expressed as dry mass fraction (%), which are calculated according to Formulas (1)–(3):(1)$$\mathrm{TPs}=\frac{A_{1}\times 1.957\times 2}{1000}\times \frac{V_{1}}{V_{2}\times m\times \omega }\times 100$$
(2)$$\mathrm{FAA}=\frac{C_{1}/1000\times V_{1}/V_{2}}{m\times \omega }\times 100\%$$
(3)$$\mathrm{CAF}=\frac{C_{2}\times V_{1}/1000\times 100/10\times 50/25}{m\times \omega }\times 100\%$$
where *V*_1_ is the total amount of TS (mL), *V*_2_ is the amount of TS for determination (mL), and *m* is the mass of the sample (g), *ω* is the dry matter content (%) of the sample, *C*_1_ is the mass of theanine found by *A*_2_ from the FAA standard curve, and *C*_2_ is the corresponding content (mg/mL) found by *A*_3_ from the CAF standard curve.

#### 2.2.2. Acquisition and Correction of Hyperspectral Data

Hyperspectral data acquisition and correction are carried out according to the methods of Chen et al. [29], the basic acquisition and analysis process is shown in Figure 1. The hyperspectral imaging acquisition system includes one imaging spectral camera, four symmetrically distributed halogen linear light sources (hsia-ls-t-200w, China), computers, and other components. The outside of the whole acquisition system is closed by a black, dark box. In addition, the hyperspectral camera has 1101 × 960 (space × spectral) pixels, the spectral range of the collected image is in the visible near-infrared bands (391–1010 nm), and the reflectivity of 360 bands can be measured.

In order to avoid the influence of dark current inside the spectral camera and improve the signal-to-noise ratio of hyperspectral image, the method mentioned in Talens’s [30] article was referred to for black-and-white correction of the collected original hyperspectral images (*R*_0_). That is, before collecting the sample image, collect the standard whiteboard to obtain the white reference image (*W*), then turn off the power, screw on the lens cover to collect the black reference image (*B*), and then use Equation (4) to calculate the reflectance image (*R*). Wherein, 65,552 in Formula (4) is the maximum value of digital number (DN).
(4)$$R=65,552\times \frac{R_{0}-B}{W-B}$$

### 2.3. Data Acquisition

Firstly, the hyperspectral images after black-and-white correction were standardized. The hyperspectral images were opened in the image processing software Spec View (Dualix Spectral Imaging, Chengdu, China) and corrected by using the analysis tool lens calibration and reflectivity calibration.

Secondly, spectral variables were extracted through ENVI5.3 (Research System Inc., Boulder, CO, USA). Open the corrected hyperspectral image in the ENVI5.3, select the image of the whole tea sample as the region of interest (ROI), extract the average reflection spectrum value of the sample, and then obtain the spectral reflection curve of the sample. A total of 116 × 360 (number of samples × number of variables) spectral matrices were obtained.

### 2.4. Preprocessing Methods of Spectral Data

Due to the influence of hyperspectral acquisition instruments or environmental factors, the original spectra of tea leaves have problems, such as scattering effect, random noise, and system noise, which will weaken the spectral signal of tea biochemical contents and is not conducive to the establishment of regression model. Therefore, before modeling, three preprocessing algorithms, MSC, S-G, and 1D, were combined to preprocess the original spectral data of tea leaves.

In order to eliminate artifacts or defect spectra in the data matrix, MSC algorithm was used to make each spectrum closer to some “ideal” spectra. In order to obtain the best estimate of spectral data points and effectively reduce the random noise of average reflection spectrum, S-G was used to “average” or “fit” each point within a certain width window of single point spectral data. In order to eliminate baseline drift and separate overlapping spectral peaks, 1D was used to enhance a small amount of information in the spectrum and estimate the difference between two subsequent spectral data points. Among them, the algorithm formula of differential method 1D is shown in Equation (5).
(5)$$\frac{d_{y}}{d_{\lambda }}=\frac{y_{i+1}-y_{i}}{\Delta \lambda }$$
where *y* is the spectrum absorbance, *λ* is the wavelength, *y_i_* is the spectrum of the *i* th sample, Δ*λ* is the wavelength interval.

### 2.5. Feature Band Extraction

In this study, spectral data of 360 bands were gained in the spectral range of 391–1010 nm. In order to improve the efficiency of later modeling, SPA, CARS, and UVE algorithms were used to select the representative bands as “feature bands” from all spectral data and eliminate the bands that are not useful for this study so as to reduce the amount of data calculation. Among them, the wavelength with the least redundancy of spectral information was selected by the SPA algorithm to solve the collinear problem [31]. The wavelength point with the larger absolute value of the regression coefficient was selected by the CARS algorithm to effectively find the best spectral combination [32]. The complexity of spectral data was reduced by UVE algorithm to improve model effect [33].

### 2.6. Model Construction and Accuracy Verification

Three machine learning methods, SVM, PLS, and RF, were used to construct the regression model between the spectral data of tea samples and their quality components.

In the evaluation system of the model, the R^2^, root mean square error (RMSE) and relative analysis error (RPD) were used to express the effect of the prediction model. Among them, the higher the R^2^ value, the closer it is to 1, indicating the higher the accuracy of the model. On the contrary, the lower the RMSE value, the closer it is to 0, indicating the higher the accuracy of the model [34]. The larger the RPD value, the more reliable the model is. If the RPD value is less than 1.4, it indicates poor prediction performance, and if the RPD value is greater than 1.4, it indicates that it can be used for model analysis [35].

In this study, the model establishment and accuracy verification were carried out by MATLAB (The Math Works, Natick, MA, USA).

## 3. Results and Discussion

### 3.1. Analysis of Quality Components

Standard curves of FAA and CAF were developed, and linear equations and R^2^ were calculated (Table 1). It can be seen from the data in Table 1 that the linear R^2^ of the standard curves of FAA and CAF are both greater than 0.99, indicating a good linear relationship, which can be used as the correction curve for the determination of the total amount of FAA and CAF.

Based on this, the content of the TPs, FAA, and CAF components in the tea samples were analyzed during withering and fermentation (Figure 2). The results showed that with the extension of the withering time, the change in the TPs content was unstable, gradually decreased from 1 to 16 h, and had no significant change after 16 h. The content of the FAA increased steadily from 1 to 16 h, reached the maximum value at 16 h, and then had no obvious change. The content of the CAF did not change much (Figure 2a). During the fermentation process, the content of the TPs decreased sharply in 1–3 h, reached the lowest value at 3 h, and then did not change. However, the content of the FAA and CAF did not change significantly during the fermentation process (Figure 2b).

Thus, it is suggested to judge the degree of withering by using the change in the FAA content. When the FAA content is stably distributed in a certain range, it is considered as moderate withering. The change in the TPs content was used to judge the degree of fermentation. When the TPs content is stably distributed in a certain range, it is considered to be moderate fermentation.

### 3.2. Division of Modeling Sample Set

The data set was randomly divided into training and testing sets in the ratio of 4:1. Table 2 shows the data distribution of the training set and testing set, including the maximum, minimum, average, and standard deviation. The results show that the content of the quality components in the samples varies greatly, indicating that the samples are well representative.

### 3.3. Preprocessing of Hyperspectral Data

In order to reduce noise interference and improve the correlation between the spectral data and the tea quality components, the MSC, 1D, and S-G algorithms were used to preprocess the hyperspectral data (Figure 3). The results show that compared with the original spectrum, the spectral curve after the combined pretreatment of MSC, 1D, and S-G is more stable, the peaks and troughs are more prominent, and the resolution and sensitivity of the spectrum are improved.

### 3.4. Selection of Characteristic Bands

Although the hyperspectral data have the characteristics of a wide spectral range and continuous wavelength, with the increase in the bands and samples, there will be problems such as band collinearity and data redundancy [36]. The selection of the characteristic band is one of the important analysis methods of hyperspectral imaging technology, which is conducive to simplifying the complexity of the prediction model and improving the accuracy of the prediction model [37,38].

In this study, the SPA, CARS, and UVE algorithms were used to select the characteristic bands (Figure 4, Table 3). The results show that among the selection methods of the characteristic bands of the TPs content, the number of characteristic bands selected by UVE is the most, which is 159, and the number of the characteristic bands selected by the SPA is the least, which is 13. Among the selection methods of the characteristic bands of the FAA content, the number of the characteristic bands selected by UVE is the most, 174, and the number of the characteristic bands selected by the SPA is the least, 16. Among the selection methods of the characteristic bands of the CAF content, the number of the characteristic bands selected by UVE is the most, which is 90, and the number of the characteristic bands selected by the CARS is the least, which is 13.

In brief, the characteristic bands screened by the UVE algorithm are the most, and the bands screened by the SPA and CARS algorithms are less. By modeling the selected characteristic variables, it is found that the prediction accuracy of the model based on the characteristic bands selected by the CARS or SPA algorithm is higher, while the prediction accuracy of the model based on more characteristic bands selected by the UVE algorithm is the lowest. Many reports indicated that selecting some important spectral variables represents better prediction outcomes than spectra containing redundant variables [39,40]. For example, by comparing the models using the CARS algorithm for processed and unprocessed spectral data, Yang et al. found that the characteristic wavelength model established after screening was better than the full wavelength model [25].

Therefore, selecting the characteristic wavelength is an important step in processing a large amount of spectral data of hyperspectral images. Through this step, the amount of data can be reduced, and a prediction model with stronger generalization ability can be obtained.

### 3.5. Establishment and Analysis of Model

The SVM, PLS, and RF machine learning methods were used to establish the regression model between the characteristic bands and the quality components content. The prediction results are shown in Table 4. The results showed that the nine models achieved excellent results, indicating that the extracted characteristic bands basically covered the characteristic information of the quality components in the withering and fermentation processes. Among them, in the TPs content prediction, the CARS-PLS model has the highest prediction accuracy, and the RP^2^, RMSEP, and RPD are 0.911, 0.003, and 5.223, respectively, while SPA-SVM has the worst prediction effect. In the prediction of the FAA content, the SPA -PLS model has the highest prediction accuracy, and the RP^2^, RMSEP, and RPD are 0.882, 0.001, and 2.974, respectively, while the UVE-RF model has the worst prediction effect. In the CAF content prediction, the CARS-PLS model has the highest prediction accuracy, and the RP^2^, RMSEP, and RPD are 0.814, 0.003, and 2.426, respectively, while the CARS-RF model has the worst prediction effect.

In general, the SVM and PLS models are the best, while the RF model is the worst. Although the RF model has the same accuracy as the other two models in the modeling stage, it performs poorly in the inversion and prediction stages. This is the same as the research results of Wang et al. They established three models, RF, SVM, and PLS, to classify and evaluate the quality of Ganoderma lucidum. The results show that compared with RF, the results of the SVM and PLS methods were more satisfying, and both of them have an accuracy of 1.000 on the test set [41]. This may be because some specific noisy data in the inversion stage may lead to the overfitting of the model, so it is difficult to predict beyond the data range of the training set. As a linear regression model, PLS can only use the linear information in the spectral data, but it can easily identify the system information and noise so as to achieve better results [12,42]. Similarly, the SVM model can make full use of the linear and nonlinear information in the spectral data and has achieved good results in the prediction of quality components [43,44].

The scatter diagram of the predicted values of the TPs, FAA, and CAF content based on the SVM, PLS, and RF models are shown in Figure 5. The solid blue line represents the ideal correlation regression line between the predicted values and the actual values of the TPs, FAA, and CAF content. From the fitting effect of the model in Figure 5, the predicted values of the samples are close to the regression line, which shows that the prediction effects of the three models have good robustness and can be used for quantitatively predicting the withering and fermentation degrees of tea raw materials.

## 4. Conclusions

The results show that it is feasible to quickly quantify the degree of the withering and fermentation of tea leaves based on hyperspectral imaging technology. The main conclusions are as follows:(1)Three methods for selecting characteristic bands, SPA, CARS, and UVE, are compared comprehensively. Among them, CARS (TPs-CARS-PLS, RP^2^ = 0.91) and SPA (TPs-SPA-PLS, RP^2^ = 0.90) achieved higher results, which not only ensures the accuracy of the model but also greatly reduces the complexity of the model.(2)Three modeling methods, SVM, PLS, and RF, are compared. The SVM (TPs-UVE-SVM, RP^2^ = 0.90) and PLS (TPs-CARS-PLS, RP^2^ = 0.91) models have strong robustness and high model accuracy. They are more suitable for the on-line monitoring of black tea quality and the intelligent judgment of the withering and fermentation degrees.(3)The inversion results of the TPs, FAA, and CAF content and hyperspectral data are compared. The prediction results of the TPs and FAA content are better. Among them, FAA-SPA-PLS (RP^2^ = 0.88) is the optimal model for judging the degree of withering, and TPs-CARS-PLS (RP^2^ = 0.91) is the optimal model for judging the degree of fermentation.

In brief, through the study of methods and models, the time required for modeling can be greatly reduced and the robustness of the model can be improved. It can also monitor the content of key components in the withering and fermentation processes of fresh tea leaves in real time, provide effective big data information, and then accurately and quickly judge the degree of withering and fermentation, which is of great significance to shorten the processing time and reduce the processing costs.

## Figures and Tables

**Figure 1 foods-11-02537-f001:**
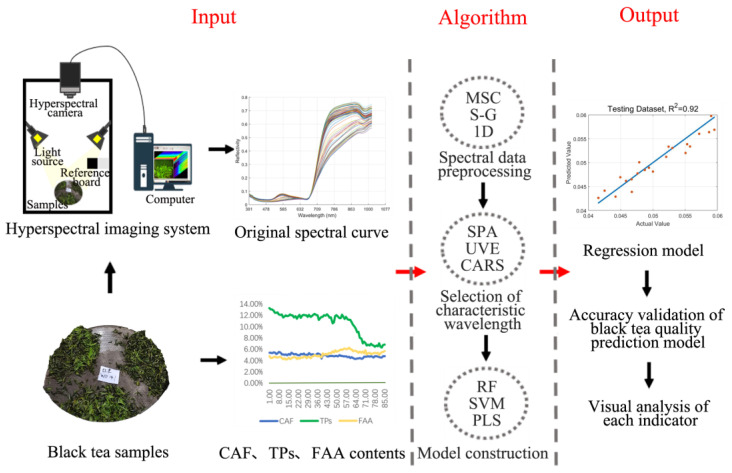
Acquisition and analysis of hyperspectral data.

**Figure 2 foods-11-02537-f002:**
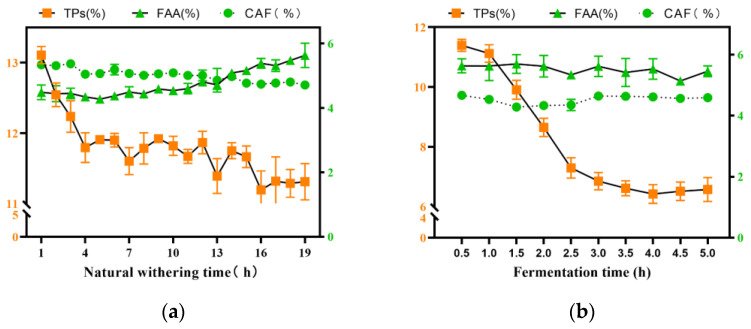
Changes of TPs, FAA, and CAF contents during withering and fermentation of fresh tea leaves. (**a**) Changes of quality components during tea withering; (**b**) changes of quality components during tea fermentation.

**Figure 3 foods-11-02537-f003:**
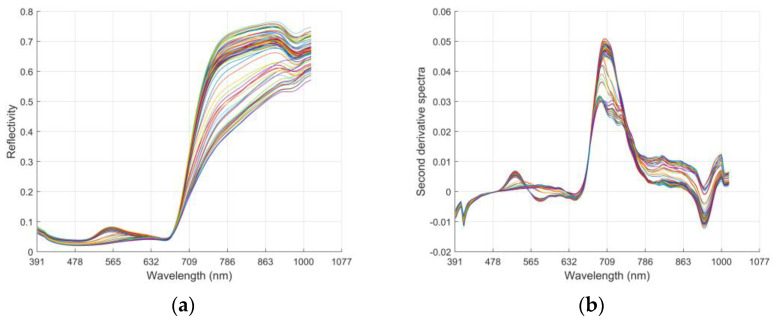
Raw data and spectra after pretreatment. (**a**) Original spectra of tea samples; (**b**) spectra after preprocessing by MSC + 1D + S-G algorithm schemes follow another format.

**Figure 4 foods-11-02537-f004:**
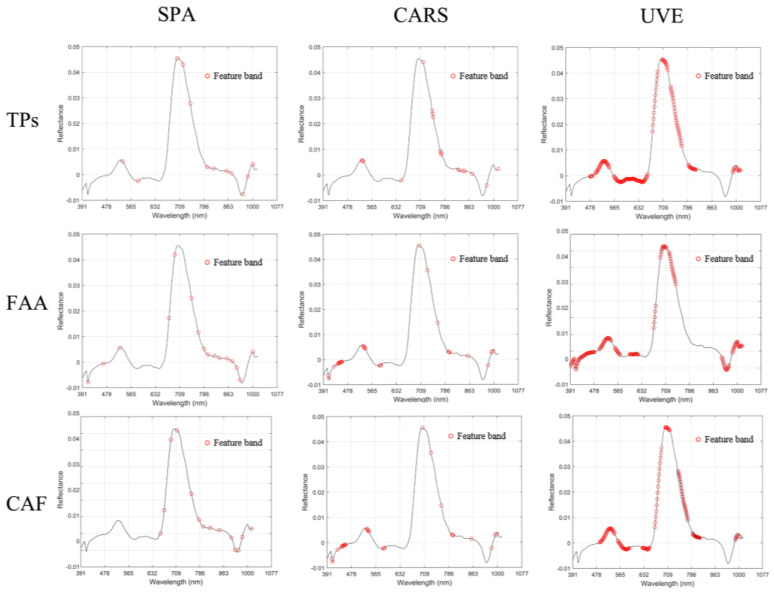
Distribution of characteristic bands.

**Figure 5 foods-11-02537-f005:**
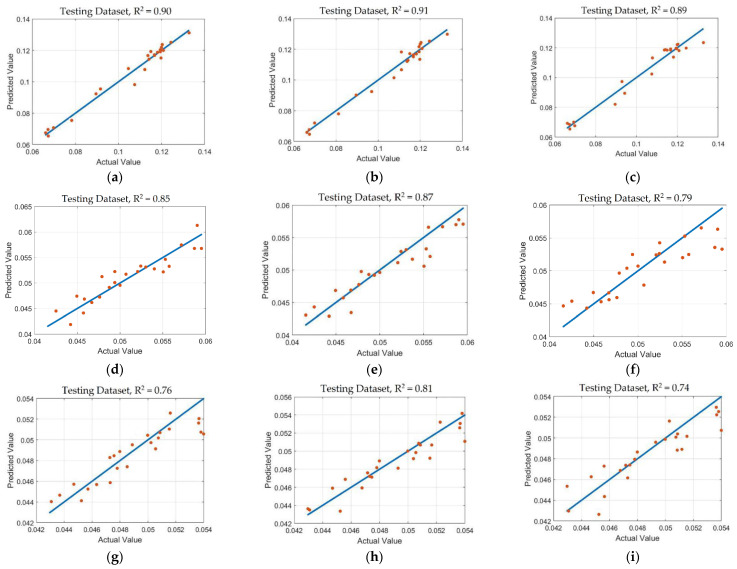
Scatter diagram of prediction of TPs, FAA, and CAF content. (**a**–**c**) TPs content prediction results obtained by CARS-SVM, CARS-PLS, and CARS-RF models; (**d**–**f**) FAA content prediction results obtained by CARS-SVM, CARS-PLS, and CARS-RF models; (**g**–**i**) CAF content prediction results obtained by CARS-SVM, CARS-PLS, and CARS-RF models.

**Table 1 foods-11-02537-t001:** Standard curve of FAA and CAF.

Standard Sample	Linear Equation	R^2^
FAA	*A*_2_ = 34.625 *C*_1_ − 0.0895	0.9983
CAF	*A*_3_ = 26.411 *C*_2_ + 0.0141	0.9903

**Table 2 foods-11-02537-t002:** The quality component content of the sample.

	Maximum/%		Minimum/%		Average/%		Standard Deviation/%	
	Training Set	Testing Set	Training Set	Testing Set	Training Set	Testing Set	Training Set	Testing Set
TPs	12.79	13.28	6.00	6.31	10.43	10.93	2.11	2.05
FAA	6.13	6.10	4.11	4.23	5.05	4.98	0.54	0.57
CAF	5.52	5.40	4.21	4.35	4.83	4.91	0.31	0.31

**Table 3 foods-11-02537-t003:** Bands screening results.

Index	Screening Method	Number of Bands	Characteristic Bands (nm)
TPs	SPA	13	512, 569, 609, 672, 714, 764, 848, 864, 898, 913, 955, 971, 992
	CARS	16	519–522, 653, 733, 764–768, 794–796, 862, 880–882, 911, 966, 1010
	UVE	159	473–475, 488–532, 554–594, 606–667, 686–703, 719–738, 750–785, 814–840, 979–986, 997
FAA	SPA	16	409, 450, 512, 701, 724, 738, 778, 807, 823, 844, 869, 896, 911, 931, 946, 992
	CARS	30	405–407, 425, 437–450, 522–529, 580–584, 715, 748, 784, 823–826, 896, 970, 984–986
	UVE	174	391–470, 488–527, 542–559, 594–623, 679–724, 734–759, 933–960, 973–1010
CAF	SPA	14	665, 679, 703, 726, 778, 807, 823, 851, 884, 929, 944, 957, 971, 1007
	CARS	13	494–498, 542, 545, 695, 710, 748, 812, 909, 922, 1007–1008
	UVE	90	483–531, 544–582, 535–655, 676–700, 715–727

**Table 4 foods-11-02537-t004:** Modeling results.

Index	Model Valuation Index	SPA			CARS			UVE		
		SVM	PLS	RF	SVM	PLS	RF	SVM	PLS	RF
TPs	RC^2^	0.911	0.923	0.924	0.926	0.926	0.920	0.919	0.931	0.924
	RMSEC	0.006	0.005	0.005	0.005	0.00	0.005	0.005	0.004	0.005
	RMSECV	0.005	0.004	0.004	0.004	0.004	0.005	0.005	0.003	0.004
	RP^2^	0.886	0.900	0.890	0.898	0.911	0.887	0.899	0.895	0.895
	RMSEP	0.004	0.003	0.004	0.003	0.003	0.004	0.003	0.003	0.003
	RPD	3.497	5.178	2.718	4.797	5.223	3.587	4.886	4.285	4.438
FAA	RC^2^	0.857	0.850	0.880	0.870	0.854	0.852	0.860	0.847	0.877
	RMSEC	0.004	0.004	0.004	0.003	0.004	0.004	0.004	0.004	0.003
	RMSECV	0.003	0.003	0.003	0.003	0.003	0.003	0.003	0.003	0.003
	RP^2^	0.802	0.882	0.830	0.846	0.866	0.788	0.800	0.778	0.743
	RMSEP	0.002	0.001	0.002	0.002	0.002	0.003	0.002	0.002	0.003
	RPD	2.547	2.974	1.857	2.864	2.522	1.609	2.368	1.798	1.579
CAF	RC^2^	0.769	0.765	0.790	0.771	0.787	0.752	0.786	0.767	0.783
	RMSEC	0.004	0.004	0.003	0.003	0.003	0.003	0.003	0.003	0.004
	RMSECV	0.004	0.004	0.004	0.004	0.004	0.004	0.004	0.004	0.004
	RP^2^	0.756	0.757	0.748	0.763	0.814	0.742	0.721	0.741	0.752
	RMSEP	0.003	0.003	0.003	0.003	0.003	0.004	0.004	0.003	0.003
	RPD	2.052	2.045	1.540	1.754	2.426	1.488	1.403	2.015	1.700

## Data Availability

The data presented in this study are available within the article.

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
