# Peer review of "Prediction of Tea Polyphenols, Free Amino Acids and Caffeine Content in Tea Leaves during Wilting and Fermentation Using Hyperspectral Imaging"

_foods, 2022, doi:10.3390/foods11162537_

Round 1

Reviewer 1 Report

1. Line 119: “During the withering process of fresh tea leaves, samples were taken once every hour, a total of 19 times.“ Was the data collected for two days?

2. Line 122: “ the hyperspectral camera is used” what are the specifications of the hyperspectral camera? Please specify them in the manuscript.

3. Line 124: “A total of 116 samples were collected and 124 each sample was repeated 4 times“ what is the variety of the tea? What stages of tea were collected in the data?

4. Section of 2.2.1 Determination of FAA, TPs, and CAF Content. The details of such steps should be specified in the manuscript.

5. The resolution of Figure 1 is low, please change it to a high-resolution one. Same for other figures as well.

6. Line 177: “In this study, we extracted 360 bands of spectral data.“ what is the spectral range of such 360 bands?

7. Did the authors have the tea sample images? If you do, please provide them as one Figure shown in the manuscript.

8. Please simplify section “4.1. It is feasible to quantitatively predict the quality components of tea leaves by hyperspectral imaging technology, and then quantitatively judge the withering and fermentation degree of tea raw materials”

Author Response

Dear reviewer, we are very grateful for your comments. Your review is very serious and has greatly improved our manuscript.

  1. Line 119: “During the withering process of fresh tea leaves, samples were taken once every hour, a total of 19 times.” Was the data collected for two days?

√Thanks for your good question. Yes, we collected data for two consecutive days. (Line 143)

  1. Line 122: “the hyperspectral camera is used” what are the specifications of the hyperspectral camera? Please specify them in the manuscript.

√Thank you for your kind advice. We have added the specification of hyperspectral camera in the manuscript. (Line 139)

  1. Line 124: “A total of 116 samples were collected and 124 each sample was repeated 4 times” what is the variety of the tea? What stages of tea were collected in the data?

√OK, thank you for your careful work. In this experiment, the raw materials of fresh tea leaves belonged to the varieties of Zhongcha, and the tenderness was one bud and one leaf. each sample was repeated for 4 times. 76 and 40 samples were collected during withering and fermentation, respectively; in total, 116 samples were collected. We have added it. (Line 133-135 and 141-143)

  1. Section of 2.2.1 Determination of FAA, TPs, and CAF Content. The details of such steps should be specified in the manuscript.

√OK, thank you for your careful work. We have specified it. (Line 150-175)

  1. The resolution of Figure 1 is low, please change it to a high-resolution one. Same for other figures as well.

√Many thanks for your careful work. We have corrected it. (Figure 1-5)

  1. Line 177: “In this study, we extracted 360 bands of spectral data.“ what is the spectral range of such 360 bands?

√OK, thank you. 360 bands of spectral data in the 391-1 010 nm spectral. We have added it. (Line 227)

  1. Did the authors have the tea sample images? If you do, please provide them as one Figure shown in the manuscript.

√OK, thank you. We have the tea sample images, but they have been placed on the left side of Figure 1.

  1. Please simplify section “4.1. It is feasible to quantitatively predict the quality components of tea leaves by hyperspectral imaging technology, and then quantitatively judge the withering and fermentation degree of tea raw materials”.

√Thank you for your kind advice. This section has been simplified and unimportant sentences have been deleted. (Line 344-391)

Reviewer 2 Report

-The manuscript evaluated the contents of tea polyphenols, free amino acids, and caffeine during withering and fermentation processing by using hyperspectral imaging. The novelty needs to be clarified and the title seems not to properly describe the whole manuscript, which needs to be modified. The specific comments are as follows:

Line 99: the authors stated "there are few reports on application of hyperspectral imaging technology in black tea processing", please refer to the following references:  "Quantitative prediction and visualization of key physical and chemical components in black tea fermentation using hyperspectral imaging", "Hyperspectral imaging technology coupled with human sensory information to evaluate the fermentation degree of black tea", "High-sensitivity hyperspectral coupled self-assembled nanoporphyrin sensor for monitoring black tea fermentation", "Classification of five Chinese tea categories with different fermentation degrees using visible and near-infrared hyperspectral imaging", "Nondestructive Testing and Visualization of Catechin Content in Black Tea Fermentation Using Hyperspectral Imaging" and so on, I don't think it is few (means 0-1) reports the application of hyperspectral imaging in black tea processing. Please state the novelty of your study.

Line 127: please provide the detailed procedures for how you determined the FAA, TPS, and CAF content. 

Line 147: I don't understand "the digital quantization value 65552". Please explain.

Line 175: Please explain what is the meaning of "dy, yi+1" and so on in equation 2.

Line 193: Please combine the sections of results and discussion to better understand.

Table 2: why do the characteristic bands chosen by UVE have two different bands? It is nearly the whole range and not the specific feather band. What standard do you judge that is feather band? It is hard to understand the meaning of Table 2.

Line 277-289: it should be in the introduction.

The conclusion is too long and should be concise. Please shorten it and keep the main finding.

Brackets: all the brackets should be in an English style, please change them all.

Figures: The quality and resolution of all the figures are so low and very hard to read, please improve all of them.

P.s. it is usually to avoid the subjective opinion in the academic paper, therefore, all the sentences should be passive tense and avoid using the subjective words like "we", please revise throughout the whole manuscript. 

Author Response

Dear reviewer, we are very grateful for your comments. Your review is very serious and has greatly improved our manuscript.

The manuscript evaluated the contents of tea polyphenols, free amino acids, and caffeine during withering and fermentation processing by using hyperspectral imaging. The novelty needs to be clarified and the title seems not to properly describe the whole manuscript, which needs to be modified.

√Thank you for your kind advice. We have changed the title to properly describe the whole manuscript. (Line 2-4)

The specific comments are as follows:

  1. Line 99: the authors stated "there are few reports on application of hyperspectral imaging technology in black tea processing", please refer to the following references: "Quantitative prediction and visualization of key physical and chemical components in black tea fermentation using hyperspectral imaging", "Hyperspectral imaging technology coupled with human sensory information to evaluate the fermentation degree of black tea", "High-sensitivity hyperspectral coupled self-assembled nanoporphyrin sensor for monitoring black tea fermentation", "Classification of five Chinese tea categories with different fermentation degrees using visible and near-infrared hyperspectral imaging", "Nondestructive Testing and Visualization of Catechin Content in Black Tea Fermentation Using Hyperspectral Imaging" and so on, I don't think it is few (means 0-1) reports the application of hyperspectral imaging in black tea processing. Please state the novelty of your study.

√OK, thank you for your kind advice. We have revised this part, restated the novelty of this study, and referred to the literature: "Quantitative prediction and visualization of key physical and chemical components in black tea fermentation using hyperspectral imaging" and "Nondestructive Testing and Visualization of Catechin Content in Black Tea Fermentation Using Hyperspectral Imaging". (Line 101-113)

2.Line 127: please provide the detailed procedures for how you determined the FAA, TPS, and CAF content.

√OK, thank you for your careful work. We have specified it. (Line 150-175)

3.Line 147: I don't understand "the digital quantization value 65552". Please explain.

√OK, thank you! Digital number (DN) is a representation of pixel value, which has no unit and no practical significance. In this study, the DN threshold of hyperspectral data is [0-65552], and 65552 used in formula 4 in the manuscript is the maximum value of DN. (Formula 4)

4.Line 175: Please explain what is the meaning of "dy, yi+1" and so on in equation 2.

√OK, thank you! We added the meaning of "dy, yi+1" and so on to equation 2. (Line 224 and 225)

5.Line 193: Please combine the sections of results and discussion to better understand.

√OK, thank you for your good advice. In order to better understand, we modified the discussion part and fully combined the relevant test results.

6.Table 2: why do the characteristic bands chosen by UVE have two different bands? It is nearly the whole range and not the specific feather band. What standard do you judge that is feather band? It is hard to understand the meaning of Table 2.

√OK, thank you for your careful work. We are very sorry for the typographical errors on the Table 2. We have modified them.

7.Line 277-289: it should be in the introduction.

√OK, thank you for your valuable advice. We have moved or deleted it. (Line 52-54, 92-93 and 342-356)

8.The conclusion is too long and should be concise. Please shorten it and keep the main finding.

√Thank you for your kind advice. These unimportant sentences have been deleted. (Line 439-470)

9.Brackets: all the brackets should be in an English style, please change them all.

√Thank you! We have checked and modified.

10.Figures: The quality and resolution of all the figures are so low and very hard to read, please improve all of them.

√Many thanks for your careful work. We have improved all of them. (Figure 1-5)

11.P.s. it is usually to avoid the subjective opinion in the academic paper, therefore, all the sentences should be passive tense and avoid using the subjective words like "we", please revise throughout the whole manuscript.

√Thank you! We have carefully checked and revised the manuscript! (Line 117, 186, 198, 213-221, 227-236, 259, 265, 281, 289 and 403)

Reviewer 3 Report

Dear Authors

In this research work monitoring model of biochemical components of tea leaves based on hyperspectral imaging technology was established to quantitatively judge the withering degree and fermentation degree of fresh tea leaves. However, I can send some details to improve the manuscript:

Materials and Methods

Figure 1. Image quality is poor. The small letters cannot be distinguished.

Results

Figure 2. Image quality is poor. The small letters cannot be distinguished.

Figure 3. Image quality is poor. The small letters cannot be distinguished.

Figure 4. Image quality is poor. The small letters cannot be distinguished.

Figure 4. The small letters cannot be distinguished. I recommend that you leave only two decimal places for the values that appear on the "X" and "Y" axis.

Author Response

Dear reviewer, we are very grateful for your comments. Your review is very serious and has greatly improved our manuscript.

Materials and Methods

1.Figure 1. Image quality is poor. The small letters cannot be distinguished.

√OK, thank you for your careful work. We have corrected it. (Figure 1)

Results

2.Figure 2. Image quality is poor. The small letters cannot be distinguished.

√Thank you! We have corrected it. (Figure 2)

3.Figure 3. Image quality is poor. The small letters cannot be distinguished.

√Thank you! We have corrected it. (Figure 3)

4.Figure 4. Image quality is poor. The small letters cannot be distinguished.

√Thank you! We have corrected it. (Figure 4)

5.Figure 4. The small letters cannot be distinguished. I recommend that you leave only two decimal places for the values that appear on the "X" and "Y" axis.

√OK, thank you for your careful work. We have changed the information of coordinate axis.

Round 2

Reviewer 2 Report

The authors addressed most of my comments, there is still some points need to be improved:

Line 161: "Determination of CAF" should be in the new paragraph. 

Line 245 and Line 336: As the authors agreed to combine the results and discussion parts, why is there still a discussion section? All (Lines 337-443) should be integrated in Line 245 and the section should be renamed as "Results and Discussion" 

For the comment "Table 2: why do the characteristic bands chosen by UVE have two different bands? It is nearly the whole range and not the specific feather band. What standard do you judge that is feather band? It is hard to understand the meaning of Table 2." The author did not answer my question about how the authors selected the characteristic bands, based on what method? In the material and method (Line 227-231), this was not clearly stated either. 

Author Response

Dear reviewer,

Thank you very much for your time involved in reviewing the manuscript and your very encouraging comments on the merits.

Comments:

The authors addressed most of my comments, there is still some points need to be improved:

1.Line 161: "Determination of CAF" should be in the new paragraph.

√Thank you for your careful work. We have corrected it. (Line 157,161,165)

2.Line 245 and Line 336: As the authors agreed to combine the results and discussion parts, why is there still a discussion section? All (Lines 337-443) should be integrated in Line 245 and the section should be renamed as "Results and Discussion"

√Thanks for your great suggestion on improving the accessibility of our manuscript. Sorry, we have misunderstood "combination". We have updated the "Results" and "Discussion" sections to "Results and Discussion". (Lines 255-378)

3.For the comment "Table 2: why do the characteristic bands chosen by UVE have two different bands? It is nearly the whole range and not the specific feather band. What standard do you judge that is feather band? It is hard to understand the meaning of Table 2." The author did not answer my question about how the authors selected the characteristic bands, based on what method? In the material and method (Line 227-231), this was not clearly stated either.

√Thank you for the detailed review. UVE, CARS and SPA algorithms are used to screen the characteristic bands. We have added the suggested content to the manuscript on lines 236-241. In fact, the UVE algorithm selects three bands with different ranges, corresponding to three biochemical components (TPs, FAA, CAF). We have supplemented Table 2. Please refer to table 2. The same is true for CARS and SPA algorithms. In addition, the characteristics of the UVE algorithm determine that the selected bands are almost the whole range. The standard for judging feather bands is the characteristic bands screened by these three algorithms.